# Normalizing Untargeted Periconceptional Urinary Metabolomics Data: A Comparison of Approaches

**DOI:** 10.3390/metabo9100198

**Published:** 2019-09-21

**Authors:** Ana K. Rosen Vollmar, Nicholas J. W. Rattray, Yuping Cai, Álvaro J. Santos-Neto, Nicole C. Deziel, Anne Marie Z. Jukic, Caroline H. Johnson

**Affiliations:** 1Department of Environmental Health Sciences, Yale School of Public Health, New Haven, CT 06510, USA; ana.rosenvollmar@yale.edu (A.K.R.V.); nicholas.rattray@strath.ac.uk (N.J.W.R.); ping.cai@yale.edu (Y.C.); alvarojsn@iqsc.usp.br (Á.J.S.-N.); nicole.deziel@yale.edu (N.C.D.); 2Strathclyde Institute of Pharmacy and Biomedical Sciences, University of Strathclyde, G4 0RE Glasgow, UK; 3São Carlos Institute of Chemistry, University of São Paulo, São Carlos 13566-590, SP, Brazil; 4Epidemiology Branch, National Institute of Environmental Health Sciences, Durham, NC 27709, USA; jukica@niehs.nih.gov

**Keywords:** urinary dilution, normalization, pregnancy, creatinine, specific gravity, probabilistic quotient normalization

## Abstract

Metabolomics studies of the early-life exposome often use maternal urine specimens to investigate critical developmental windows, including the periconceptional period and early pregnancy. During these windows changes in kidney function can impact urine concentration. This makes accounting for differential urinary dilution across samples challenging. Because there is no consensus on the ideal normalization approach for urinary metabolomics data, this study’s objective was to determine the optimal post-analytical normalization approach for untargeted metabolomics analysis from a periconceptional cohort of 45 women. Urine samples consisted of 90 paired pre- and post-implantation samples. After untargeted mass spectrometry-based metabolomics analysis, we systematically compared the performance of three common approaches to adjust for urinary dilution—creatinine adjustment, specific gravity adjustment, and probabilistic quotient normalization (PQN)—using unsupervised principal components analysis, relative standard deviation (RSD) of pooled quality control samples, and orthogonal partial least-squares discriminant analysis (OPLS-DA). Results showed that creatinine adjustment is not a reliable approach to normalize urinary periconceptional metabolomics data. Either specific gravity or PQN are more reliable methods to adjust for urinary concentration, with tighter quality control sample clustering, lower RSD, and better OPLS-DA performance compared to creatinine adjustment. These findings have implications for metabolomics analyses on urine samples taken around the time of conception and in contexts where kidney function may be altered.

## 1. Introduction

With growing interest in the exposome—the totality of an individual’s exposures across the life course—metabolomics is a powerful tool for measuring both chemical exposures and biological responses at a molecular level [1]. Because of the challenges in continuous longitudinal exposome monitoring, efforts to characterize the exposome often focus on critical developmental windows, including the periconceptional period and pregnancy [2,3]. In environmental health research, urine is a valuable biospecimen not only for evaluating the presence of environmental chemicals through biomonitoring, but also for assessing the biological impact of exposures through changes to endogenous metabolites. However, accounting for differential urinary dilution across samples through application of a normalization strategy is essential to reduce measurement error and accurately interpret findings [4]. Although urinary creatinine concentration is commonly used to adjust for urinary dilution in biomonitoring studies, creatinine excretion is known to vary based on multiple factors, including sex, age, diet, exercise, muscle mass, body mass index (BMI), and health conditions that impact kidney function [4,5,6,7,8,9,10]. Normal pregnancy substantially alters kidney function and creatinine excretion, with changes documented as early as 3–6 weeks after the last menstrual period (LMP) [10], and clinically observable through increased urine production and decreased urinary creatinine concentrations [11]. Hormonal changes of pregnancy promote fluid and electrolyte retention, with concomitant vasodilation to accommodate the increase in water and blood volume [12]. Kidney volume increases 30% and glomerular filtration rate increases 50–80% above pre-pregnancy levels by the end of the first trimester [13,14]. Using creatinine to adjust for urinary concentration during times of changing kidney function, such as the critical window of early pregnancy, has the potential to introduce confounding and measurement errors [5,8,15,16].

In metabolomics research, there is no consensus on the optimal normalization strategy to account for concentration variability in urinary metabolomics data, especially in situations where creatinine excretion or kidney function may be impacted [17,18,19]. Metabolomics normalization strategies adjust data for both analytical and biological variability: analytical variability results from changes in instrument performance over the course of an experiment [20,21], while biological variability is due to biological differences between subjects, including sample concentration [17,18]. Additional variability also can be introduced by experimental conditions and study design decisions. Normalization can occur pre-acquisition, in which samples are diluted to a common concentration before analysis, or post-acquisition, which adjusts the data after analysis [22]. Other common approaches used in metabolomics include normalization to an internal standard or reference compound such as creatinine, and adjustment using specific gravity or osmolality. Specific gravity is a ratio of the density of urine (in this case) to pure water at constant temperature, while osmolality measures the solute concentration in a given solution in moles per kilogram, or osmoles [23]. Additionally, there are exclusively statistical approaches developed specifically for metabolomics research, such as probabilistic quotient normalization (PQN) [24,25]. These methods typically use a summary statistic of metabolite responses for normalization. PQN uses the distribution of metabolites across pooled quality control (QC) samples as a normalization factor, and is a fundamentally different approach to normalization from creatinine or specific gravity adjustment because it relies on the attributes of QC samples as a reference, rather than on the study samples themselves. First developed for nuclear magnetic resonance (NMR) spectroscopy [24], PQN has since been applied widely within mass spectrometry-based metabolomics [25].

Most studies systematically comparing post-acquisition normalization approaches for untargeted urinary metabolomics data have just compared statistical approaches [22,25,26,27]. Only two studies have compared both statistical approaches and methods that incorporate direct measurements from samples, such as using an internal standard, creatinine, or specific gravity to adjust for concentration [22,28]. Although pre-acquisition normalization strategies have yielded promising results [19,28,29], application of this strategy can be impractical in the large cohorts increasingly common in exposome and environmental health studies, given the intensive sample preparation processes. Thus, the objective of this study was to determine the optimal post-analytical normalization approach for untargeted metabolomics analysis of periconceptional urine samples. We used a subset of women from a periconceptional cohort and compared the performance of three common approaches to adjust for urinary dilution: creatinine, specific gravity, and PQN.

## 2. Results

To compare normalization approaches, we used 90 paired pre- and post-implantation pooled urine samples from 45 women. After mass spectrometry analysis and data processing, we used support vector regression (SVR) to correct for analytical variability in all datasets. We then compared three approaches to adjust for metabolite concentration variability—creatinine, specific gravity, and PQN—using unsupervised principal components analysis (PCA), and the relative standard deviation (RSD) of pooled QC samples. Orthogonal partial least-squares discriminant analysis (OPLS-DA) was used to assess the variance in the data associated with pre- and post-implantation status for each normalization method, and models were validated with a permutation test. Statistically significant changes in peaks from pre- to post-implantation for each normalization approach were evaluated with the Wilcoxon paired signed-rank test.

PCA scores plots of the raw data without application of normalization methods for analytical or biological variability show that the QC samples are not well-clustered (Figure 1 and Figure 2, panel A). PCA plots were assessed using visual inspection of the relative dispersion of QC samples [22,30]; tightly clustered QC samples are indicative of good reproducibility, as theoretically the QC samples should be in an identical position within the plot. Data normalized for analytical variability alone using SVR demonstrate tight QC clustering. When the data are further adjusted for concentration variability, QC clustering is tightest for specific gravity normalization, followed by PQN, and then creatinine. Creatinine normalization shows poor QC clustering especially for reversed-phase liquid chromatography (RPLC) data (Figure 1, panel C).

Comparison of the RSD across QC samples by normalization method (Table 1) shows that adjusting for analytical variability alone with SVR results in 94.0% (RPLC data) and 80.3% (hydrophilic interaction chromatography, HILIC, data) of peaks having an RSD < 0.3, which is a substantial improvement over the raw data. RSD is a measure of the variability of the peak area across QC samples, with a lower RSD indicating less variability and more similarity across the QC samples, which again theoretically should be identical. There is general consensus in the metabolomics research community that an RSD < 0.3 is indicative of good reproducibility [31]. Additional adjustment for concentration variability using creatinine and specific gravity normalization results in a similar RSD for both RPLC and HILIC data compared to SVR normalization. However, PQN normalization yields a substantially lower median RSD (0.08 for RPLC and 0.11 for HILIC) compared to other methods, even though when visualized via PCA plot the QCs are slightly more dispersed compared to specific gravity-normalized data (Figure 1 and Figure 2). These low RSD results are an artifact of the PQN normalization process: PQN-normalized QCs have a peak area very close to 1 because the PQN method divides the QC peak area by the median of the mean QC peak area for a given m/z, and the given QC peak area. Because normalized QC peak areas are close to 1, the standard deviation is small, resulting in an even smaller RSD. Therefore, using RSD alone to compare PQN-normalized QCs with QCs normalized by other methods may not be reliable.

We then compared the mean difference in RSD between each normalization method using paired *t*-tests (Table 2). Positive mean differences indicate the mean RSD of the first method listed in Table 2 is larger and therefore less optimal than that of the second method. These comparisons confirmed that additional normalization for urinary concentration using PQN improves the RSD compared to creatinine and specific gravity for both RPLC and HILIC data. Specific gravity performed better than creatinine for RPLC data, but there was no statistically significant difference in RSD between specific gravity and creatinine for HILIC data. The equivalent RSD between SVR and specific gravity is a result of the specific gravity normalization method, in which QC peak areas are the same as those normalized by SVR alone, leading to no difference in RSD when comparing these methods (see Section 4.6, Equation (1)).

We carried out OPLS-DA analysis to determine the variance in the data associated with pre- and post-implantation status, as well as the predictability of the model (Table 3), including only those peaks with RSD < 0.3. As a result of using this RSD threshold, PQN data included more features than any other approach because of the low RSD associated with the PQN process. The variation in the data related to pre- and post-implantation status was extremely similar for specific gravity and creatinine for RPLC data (R^2^X = 0.36 and 0.37 respectively), and much lower for PQN (R^2^X = 0.18). The low R^2^X of PQN may be related to the fact that more peaks are included that might not have relevance to implantation status because of the method’s low RSD from the PQN calculation process, which could be a source of bias. For HILIC data, it was highest for creatinine (R^2^X = 0.37), with PQN and specific gravity much lower (R^2^X = 0.25 and 0.20 respectively). For all methods, there was good group separation by implantation status (R^2^Y > 0.80 for all RPLC data, and R^2^Y > 0.75 for all HILIC data). Best group separation by implantation status was observed in PQN-normalized data, followed by specific gravity and then creatinine. The overall predictive performance of the model was similar across normalization methods for RPLC data, with the creatinine-adjusted model performing the poorest (Q^2^ = 0.58) and PQN the best (Q^2^ = 0.69). For HILIC data, the creatinine-adjusted model had a substantially lower Q^2^ (0.27) compared to specific gravity or PQN (Q^2^ = 0.53 and 0.52 respectively).

This suggests that for both RPLC and HILIC modes, models that use features that have been creatinine-adjusted result in poorer predictions for features correlated with implantation status. The difference between R^2^Y and Q^2^ is < 0.3 for all three normalization approaches in RPLC data; however, it is > 0.3 for all three approaches in the HILIC data, suggesting overfitting of the model [32]. Results of the permutation test (Appendix A, Appendix A) to validate the model were similar across all normalization methods, and in all cases had a Q^2^ intercept close to 0 (range of −0.08 to 0.09), showing reasonable model performance. For RPLC data, creatinine, specific gravity, and PQN all had a similar number of features with variable importance in projection or VIP > 1 and *q* < 0.05 (Table 3). Features meeting both of these criteria are of higher importance in the OPLS-DA model and show statistically significant differences from pre- to post-implantation. For HILIC data, PQN had the most features meeting these criteria (*n* = 2578), followed by specific gravity (*n* = 2358) and then creatinine (*n* = 1491).

Taken together, these analyses indicate that PQN and specific gravity normalization both outperform creatinine adjustment; in some tests PQN performs better than specific gravity, while in others specific gravity outperforms PQN. Specific gravity has tighter clustering of QC samples in unsupervised PCA plots compared to PQN (Figure 1 and Figure 2), and while PQN has a lower RSD than specific gravity, this is an artifact of the PQN normalization process, suggesting the two approaches perform similarly when evaluated by RSD (Table 1 and Table 2). In OPLS-DA results (Table 3) for RPLC data, PQN has a high Q^2^ but a low R^2^X, indicating good group separation but relatively little variability explained by implantation status, while specific gravity performs consistently well across assessments. For HILIC data, OPLS-DA results for PQN and specific gravity are similar. Finally, for RPLC data, PQN and specific gravity have a similar number of features meeting the VIP > 1 and *q* < 0.05 criteria, while for HILIC data PQN has more features meeting this threshold than does specific gravity. Creatinine is a poor choice for concentration normalization of periconceptional urinary metabolomics data, and either PQN or specific gravity are good alternative options.

## 3. Discussion

We found that either specific gravity or PQN are good approaches for post-acquisition normalization of untargeted, periconceptional urinary metabolomics data. Our findings suggest that creatinine adjustment may not be an effective normalization approach for samples collected around the time of conception and during early pregnancy. Although there is some consensus in the metabolomics literature that normalization for urinary dilution using creatinine may be inaccurate given the many biological factors that can influence kidney function and creatinine excretion, there is not agreement on the optimal normalization approach [17,18,22,33,34,35,36].

The performance of urinary creatinine compared to specific gravity adjustment for accurate biomarker measurement has been explored in biomedical and environmental health contexts, with studies finding specific gravity robust to the physiologic parameters that influence creatinine (i.e., age, sex), and to have less within-person variability [5,6,7,37,38,39]. These studies conclude that specific gravity is either better than or equivalent to creatinine when adjusting for urinary concentration. To our knowledge, no metabolomics studies have directly compared creatinine and specific gravity normalization, although osmolality and creatinine normalization have been compared. Osmolality, which measures the number of solute particles per kilogram of solution, is considered the gold standard approach to assessing urine concentration, and is a closely related measure to specific gravity [23]. Comparison of pre-acquisition normalization using osmolality and creatinine in a small (*n* = 5) group of hospital patients with unspecified conditions found osmolality to perform better [29], while two targeted metabolomics studies found post-acquisition normalization using either osmolality or creatinine did not impact a model’s predictive ability to identify patient diagnoses [35], or alter biomarker identification [36]. Together, these studies suggest that while there may be a growing consensus among biomedical and environmental health literature that specific gravity is more reliable than creatinine when adjusting for urinary dilution, there is scant research in the field of metabolomics comparing the approaches.

Similarly, while multiple studies have found PQN to be an optimal technique for normalizing untargeted urinary metabolomics data, to our knowledge none have compared it to non-statistical approaches for adjusting for concentration differences, such as creatinine or specific gravity [22,25,26,27]. In a study of how metabolomics data structures are impacted by normalization methods, Saccenti [34] found that application of a variety of normalization methods to adjust for urinary dilution, including PQN and creatinine, induced spurious correlations. Saccenti suggests that normalization to urine output (volume/unit time), a physiologic parameter, better preserves data structure than internal standard or statistical approaches. We found specific gravity and PQN to perform similarly based on the combined assessment of PCA plots, RSD scores, OPLS-DA results, and Wilcoxon paired sign-rank tests. Importantly, we also found that comparing PQN to other techniques using RSD can yield low RSDs that are an artifact of the calculation process, as QCs normalized using PQN are very close to 1, resulting in a small standard deviation and RSD. In normalization comparisons that include PQN, relying on RSD alone for evaluation could result in misleading conclusions.

The strengths of our study include that it is the first to systematically compare metabolomics normalization strategies for urinary concentration in periconceptional cohorts, using data from both RPLC and HILIC analyses. To our knowledge, no other studies have examined normalization approaches in periconceptional cohorts. Only one other study has examined normalization methods in the context of kidney function changes, focusing on kidney failure patients and comparing pre- and post-acquisition normalization methods [22]. Our study focuses on post-acquisition normalization techniques, which in many cases may be simpler and more feasible, especially for large cohort studies with hundreds or even thousands of subjects where dilution to a common concentration may be impractical. Comparison of normalization strategies in other periconceptional and pregnancy cohorts could provide valuable information on optimal strategies, with application to research efforts seeking to characterize the early life exposome.

Of note, our study has some limitations. The conclusion that specific gravity and PQN are similarly effective normalization approaches for the urinary metabolomics data generated from our cohort may not be generalizable to all periconceptional cohorts. Other methods may be optimal in different settings, in different kinds of cohorts, and at different times during pregnancy. Using specific gravity for normalization demands performing an additional analysis of each specimen. While this is less labor- and cost-intensive than pre-analytical normalization, methods that make use of data collected by mass spectrometry have distinct advantages in large cohort metabolomics studies. Because specific gravity is a measure of density, the presence of heavy molecular weight compounds such as protein or glucose in a urine sample can produce a dense solution, leading to an overestimation of specific gravity [23,39]. Beginning in mid-pregnancy at approximately 20 weeks gestation, glucose resorption by the proximal tubule of the kidneys is less effective, and increases in glomerular filtration rate can lead to protein excretion, although in normal pregnancies glucose and protein excretion do not exceed normal limits [14]. Additionally, in study populations with clinical conditions such as diabetes or hypertension, including during pregnancy, the use of specific gravity for normalization may be less reliable. Because our samples were collected at 3–6 weeks gestation, and subjects were all healthy, this is unlikely to impact our findings related to specific gravity.

Our findings suggest that during the periconceptional period, use of creatinine adjustment for post-analytical concentration normalization of urinary metabolomics data is not a reliable approach. Instead, either specific gravity normalization or PQN are good alternative approaches to adjust for urinary dilution, with application to metabolomics research during the periconceptional period, and other times of changing kidney function.

## 4. Materials and Methods

### 4.1. Study Populations

We used urine specimens from the Early Pregnancy Study (EPS), a prospective cohort study conducted in North Carolina from 1982–86 of 221 women without known fertility problems or chronic illnesses who were attempting to become pregnant [40,41]. Briefly, the participants collected daily first morning urine specimens for 6 months if they did not conceive or through 8 weeks gestation if they did conceive. Specimens were collected in polypropylene jars without preservatives and stored for up to 2 weeks in home freezers before being transported on ice to a central freezer and stored at −20 °C [42]. After completion of the original study, the samples were placed in long-term storage at −80 °C at the National Institute of Environmental Health Sciences (NIEHS). 

To compare normalization approaches, we randomly selected a subset of 45 subjects from the EPS cohort with a pregnancy resulting in live birth. This study was approved by the Yale University Institutional Review Board (ID 2000022845). Pregnancy was defined as a human chorionic gonadotropin (hCG) level of > 0.025 ng/mL for ≥ 3 consecutive days [41]. After identification of a pregnancy, the day of implantation was defined as the first day of the pregnancy where hCG was ≥ 0.01 ng/mL. Inclusion criteria were no tobacco, alcohol, or marijuana use; white; age 27 to 31; and BMI < 25. Study conceptions that resulted in multiple gestations or that ended with a pregnancy loss were excluded.

### 4.2. Pooled Urine Samples

As part of a previous research study (NCT01132924) [42], stored daily urine specimens were pooled within menstrual cycles for each individual woman in order to facilitate the analysis of exposures to rapidly metabolized environmental chemicals with high intra-individual variability. Pools were created by selecting three first-morning specimens that were approximately one week apart, and that were collected after the end of menses. Equal aliquots were then drawn from each of these three specimens (Figure 3). The first two samples comprising the pool were pre-ovulation and the third was post-ovulation, but pre-implantation if a participant conceived. Together, these pooled samples reflect an individual’s entire menstrual cycle, and are referred to as “pre-implantation.” “Post-implantation” pooled samples consisted of three specimens approximately one week apart that were collected after the day of implantation. The post-implantation samples correspond to the first 3 to 6 weeks of gestation as measured from the LMP. For all 45 subjects, pooled urine samples from the conception cycle (pre-implantation sample) and from early pregnancy (post-implantation sample) were analyzed, for a total of 90 paired urine samples from two biological states.

### 4.3. Specimen Preparation

Urine specimens were sent overnight on dry ice and then stored at −80 °C until preparation using a solid phase extraction technique [43]. We created a pooled QC sample consisting of equal aliquots from each sample to account for analytical variability, from which multiple QC samples were extracted. As these QC samples are identical in composition, any changes in data acquired from QC samples during an experiment represents analytical variability, and can be corrected using statistical techniques. All samples were run on 96-well plates consisting of 90 urine samples, 3 pooled QCs, 2 extraction blanks used to ensure the absence of contamination, and 1 internal standard. Urine (75 µL) was combined with distilled water (75 µL), and 120 µL of each sample with 5 µL internal standard was extracted using an Oasis HLB96 µElution Plate (Waters Corporation, Milford, MA, USA). This extract was used for analysis of polar metabolites by HILIC-mass spectrometry (MS). We then passed 150 µL methanol through the extraction plate, and transferred 120 µL of the eluent to the 96-well plate. This plate was used for analysis of non-polar metabolites by RPLC-MS.

### 4.4. UHPLC/MS Analysis

All specimens were analyzed by ultra high performance liquid chromatography (UHPLC) quadrupole time-of-flight (QToF) mass spectrometry (Xevo-G2-XS-QToFMS, Waters Corporation, Milford, MA, USA), with run-order randomized and QC samples included every 6 injections. For HILIC-MS, 2 µL of each sample were injected onto an Acquity UPLC BEH Amide column (1.7 µm particle size, 2.1 mm × 150 mm ID; Waters Corporation), using a mobile phase composed of A (10 mM ammonium bicarbonate in 5:95 acetonitrile:water) and B (95:5 acetonitrile:water) at 30 °C and a solvent flow rate of 300 µL/min in electrospray ionization (ESI) negative mode. The linear gradient elution began at 95% B (0–1 min), then went to 65% B (1–14 min), 40% B (14–18 min), back to 95% B (18–18.1 min), and held at 95% B to clean the column (18.1–23 min). For RPLC-MS, 2 µL of each sample were injected onto an Acquity UPLC BEH RPLC column (1.7 µm particle size, 2.1 mm × 50 mm; Waters Corporation, Milford, MA, USA), using a mobile phase composed of A (95% water, 5% methanol, and 0.1% formic acid) and B (95% water, 5% methanol, and 0.1% formic acid) at 40 °C and a flow rate of 400 µL/min in ESI positive mode. The linear gradient elution began at 95% A (0–2min), then changed to 5% A (2–12 min) and held at 5% for 30 s, back to 95% A (12.5–13 min), and held at 95% A to regenerate the column (13–14 min).

Mass calibration of the mass spectrometer followed the manufacturer’s guidelines and used leucine enkephalin (200 pg/mL), which was also used as the locking mass compound. The QToF was set to acquire at 30,000 resolution in sensitivity mode over the 25–1000 m/z range, with 1 scan per 300 ms. Desolvation gas temperature was 400 °C, desolvation gas flow 800 L/h of N2, and cone gas flow 100 L/h. The ESI source voltage was set to 3.5 V, and source temperature set at 120 °C. MassLynx software (Waters Corporation, Milford, MA, USA) was used as the operating system.

### 4.5. Data Processing

ProteoWizard (version 3.0.10158) was used to convert .RAW files to .mzML files. The XCMS package (version 3.8) running on R (version 3.4) was used for deconvolution and peak alignment [44,45,46]. The XCMS-generated peak tables were further processed with the MetCleaning [47] package in R using support vector regression (SVR) to account for analytical variability [20,21,48], and then normalized using creatinine, specific gravity, or PQN in Excel 2016. Before processing the HILIC data using MetCleaning, 95 features (out of 18,977 features) were removed from the dataset where peak area = 0 for ≥ 3 QC samples, resulting in 18,882 features in the SVR-normalized dataset (see Table 1). For the RPLC dataset, use of MetCleaning filtered 17 features (out of 12,811 features), resulting in 12,794 features in the SVR-normalized dataset (see Table 1). The data processing parameters used for XCMS and MetCleaning are included in Appendix B. Please contact the authors regarding data availability.

### 4.6. Normalization Methods

In metabolomics analyses, concentration normalization techniques may be applied before data acquisition, such as diluting all samples to the same concentration before analysis based on specific gravity or osmolality [22,28,29], or post-acquisition, such as correction using creatinine [29], specific gravity [19,28], or PQN [24,25]. We first used SVR, a non-parametric machine learning method that has been found to out-perform LOESS and robust spline alignments, to account for analytical variability with the MetCleaning R package [20,21,48]. After applying SVR to all datasets, we compared the performance of three post-acquisition normalization approaches to adjust for concentration variability: creatinine, specific gravity, and PQN. In the context of post-analytical normalization, we first carried out analytical normalization and then corrected for concentration differences. In the case of PQN and specific gravity, where QC samples are used for normalization, first accounting for instrumental bias such as signal drift ensures QC peak areas are correct before normalization of each sample. For approaches like creatinine adjustment that use measurements collected by mass spectrometry, normalization of uncorrected peak areas using an uncorrected creatinine value could introduce further error. We therefore chose to first apply analytical normalization to all datasets, and then compare the performance of methods adjusting for concentration variability.

Creatinine normalization was carried out by dividing each peak area by the peak area of creatinine for a given sample [5,6,7,49]. In the HILIC-MS dataset, creatinine was identified as m/z = 112.05132 [M-H]^−^ (mass-to-charge ratio, m/z) at rt = 277.282 s (retention time in seconds, rt). In the RPLC-MS dataset, creatinine was identified as m/z = 114.0662 [M+H]^+^ at rt = 21 s; because this peak was not picked in the initial XCMS processing, peak areas were obtained by manual integration of the raw data in MassLynx (version 4.2, Waters Corporation). Creatinine normalization was applied to the peak table resulting from MetCleaning, after SVR normalization.

Specific gravity (SG) was measured using an Atago PAL-10S Refractometer. Specific gravity normalization was carried out using the following formula:(1)$${PA}_{corr}=PA\times \frac{{SG}_{ref}-1}{SG-1}$$ where ${PA}_{corr}$ = specific gravity-corrected peak area, PA
*=* peak area, ${SG}_{ref}$ = reference specific gravity (in this case 1.020, the median and mean specific gravity of all samples), and SG = specific gravity measured in sample [5,6,38]. The difference SG−1 approximates the total dissolved urinary solids present in the sample [48]. The effect of this normalization method is to increase peak area when the sample is dilute ($SG<SG_{ref}$), and decrease peak area when the sample is concentrated ($SG>SG_{ref}$). Because specific gravity was not measured in the pooled QC samples, the median specific gravity across all samples was used for QC correction. While this could reduce variability due to measurement error of QC specific gravity measurements, because the QCs are comprised of equal aliquots from each sample, it is unlikely that this contributed to improving the reproducibility profile of specific gravity in this study. Specific gravity normalization was applied to the peak table resulting from MetCleaning, after SVR normalization.

The PQN technique uses a “gold standard” spectrum to normalize test features [24,25]. In this study, as per common practice, we used the distribution of QC sample m/z values as the gold standard spectrum. First, the mean peak area of the QC features was calculated for each m/z value. Then, the median between a given test feature peak area and the QC mean peak area was calculated. Finally, the test feature peak area was divided by the median, resulting in a normalized peak area. Features with peak area of 0 for all QC samples were removed before normalization (*n* = 2 for RPLC data, *n* = 1 for HILIC data). As with creatinine- and specific gravity-adjustment, PQN was applied to the peak table resulting from MetCleaning, after SVR normalization.

### 4.7. Statistical Comparison of Normalization Methods

To compare normalization techniques, we first used unsupervised PCA to visually inspect the clustering and relative dispersion of the QC samples [22,30]. Then, we calculated the RSD by dividing the standard deviation of the peak area by the mean peak area across QC samples as a measure of feature reproducibility across QCs, with ideal RSD < 0.3 or 30% [17,20,22,50]. Paired *t*-tests were used to compare the mean difference in RSD between each normalization method. OPLS-DA was used to determine the variance in the data associated with pre- and post-implantation status, and we included only those peaks for which RSD < 0.3. For each normalization method, we calculated R^2^X, R^2^Y, Q^2^, and the number of features where the variable importance in projection or VIP was greater than 1 [51]. A permutation test was conducted to validate the model, correlating the Q^2^ and R^2^ of the original data with the distribution of Q^2^ and R^2^ after 200 iterations randomly assigning pre- and post-implantation status [32]. The Wilcoxon paired signed-rank test was used to determine whether there was a statistically significant change in peak area for a given feature from pre- to post-implantation status, for each normalization method. The level of significance was set using a Benjamini–Hochberg false discovery rate (FDR)-corrected *p*-value (*q*-value) of *q* < 0.05 [52]. For each normalization method, we reported the number of features with both *q* < 0.05 and VIP > 1, common criteria applied to identify potential biomarkers [51]. All analyses were carried out in R (version 3.4.1).

## 5. Conclusions

In metabolomics studies using urine specimens, accounting for urinary concentration is essential. However, when a health status or outcome of interest alters kidney function, as is the case with normal pregnancy, creatinine may not be a reliable normalization approach. In this paper, we show that specific gravity and PQN are both more effective methods for normalizing urinary periconceptional metabolomics data compared to creatinine. These findings have implications for metabolomics analyses around the time of conception and in contexts where normal kidney function is altered.

## Figures and Tables

**Figure 1 metabolites-09-00198-f001:**
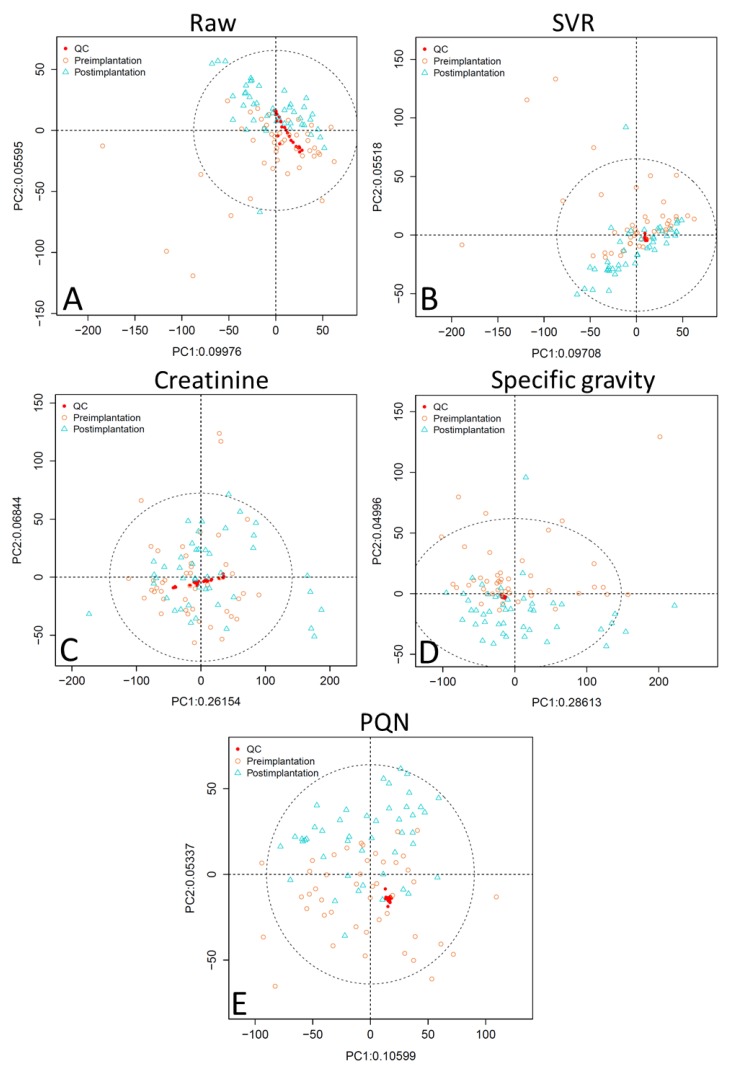
Unsupervised PCA plots comparing normalization approaches for RPLC data showing (**A**) raw data, (**B**) SVR-normalized data, (**C**) SVR and creatinine-normalized data, (**D**) SVR and specific gravity-normalized data, and (**E**) SVR and PQN-normalized data. Abbreviations: RPLC, reversed-phase liquid chromatography; SVR, support vector regression; PQN, probabilistic quotient normalization; PC1, principal component 1; PC2, principal component 2; ellipse is a 95% confidence ellipse.

**Figure 2 metabolites-09-00198-f002:**
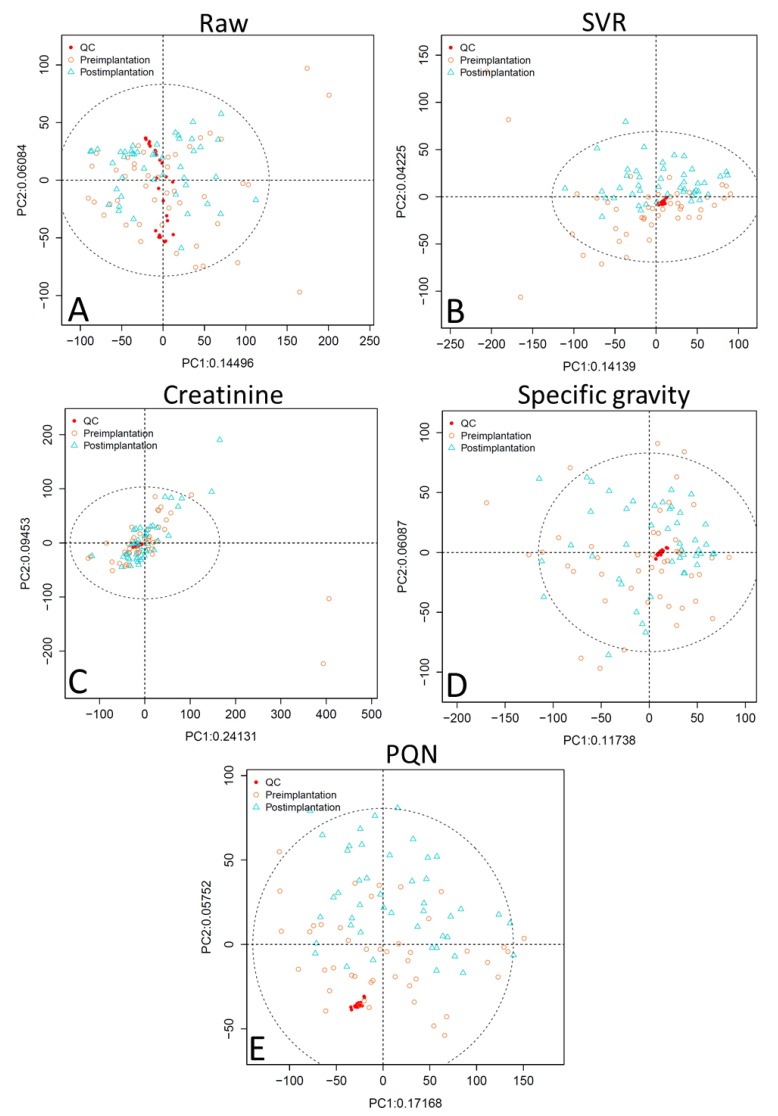
Unsupervised PCA plots comparing normalization approaches for HILIC data showing (**A**) raw data, (**B**) SVR-normalized data, (**C**) SVR and creatinine-normalized data, (**D**) SVR and specific gravity-normalized data, and (**E**) SVR and PQN-normalized data. Abbreviations: HILIC, hydrophilic interaction chromatography; SVR, support vector regression; PQN, probabilistic quotient normalization; PC1, principal component 1; PC2, principal component 2; ellipse is a 95% confidence ellipse.

**Figure 3 metabolites-09-00198-f003:**
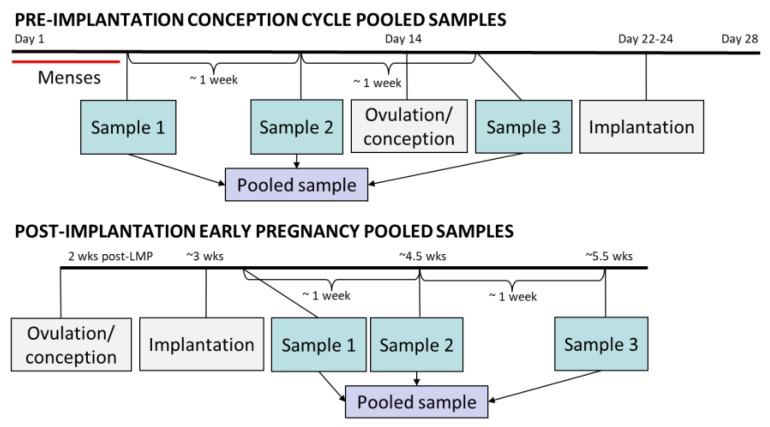
Pooled sampling strategy for pre- and post-implantation urine specimens. The pre-implantation sample depicted here is a from a theoretical conception cycle during which ovulation occurs on day 14 (the mode in this population). In most conceptions, implantation occurs approximately 8–10 days after ovulation. Gestational weeks are measured from the start of the last menstrual period (LMP), so post-implantation early pregnancy pooled samples span 3–6 weeks gestation.

**Table 1 metabolites-09-00198-t001:** Comparison of relative standard deviation of peak area for quality control samples across normalization approaches.^1^

Normalization Approach	RPLC Data		HILIC Data	
	Median RSD (IQR)	Peaks with RSD < 0.3, *n* (%)	Median RSD (IQR)	Peaks with RSD < 0.3, *n* (%)
Raw Data	0.23	9827/12,811	0.33	8271/18,977
	(0.16–0.29)	(76.7%)	(0.23–0.48)	(43.6%)
SVR	0.15	12,023/12,794	0.19	15,158/18,882
	(0.10–0.20)	(94.0%)	(0.13–0.27)	(80.3%)
SVR and Creatinine	0.18	11,744/12,794	0.20	15,104/18,882
	(0.15–0.23)	(91.8%)	(0.13–0.27)	(80.0%)
SVR and Specific Gravity	0.15	12,023/12,794	0.19	15,158/18,882
	(0.10–0.20)	(94.0%)	(0.13–0.27)	(80.3%)
SVR and PQN	0.08	12,667/12,794	0.11	18,106/18,882
	(0.05–0.11)	(99.0%)	(0.07–0.16)	(95.9%)

^1^ SVR addresses analytical variability, while creatinine, specific gravity, and PQN address concentration variability. RPLC data represent hydrophobic metabolites, and HILIC data represent hydrophilic metabolites. Raw data has been processed using XCMS without any application of a normalization technique. Abbreviations: RPLC, reversed-phase liquid chromatography; HILIC, hydrophilic interaction chromatography; RSD, relative standard deviation; IQR, interquartile range; SVR, support vector regression; PQN, probabilistic quotient normalization.

**Table 2 metabolites-09-00198-t002:** Comparison of the mean difference in relative standard deviation across quality control samples between normalization approaches using paired *t*-tests.^1^

Compared Normalization Approaches	RPLC Data		HILIC Data	
	RSD Mean Difference ^3^	*p*-value	RSD Mean Difference ^3^	*p*-value
Raw and SVR ^2^	0.100	*p* < 2.2 × 10^−16^	0.160	*p* < 2.2 × 10^−16^
SVR and Creatinine ^2,4^	−0.040	*p* < 2.2 × 10^−16^	−0.002	0.21
SVR and Specific Gravity ^2,4^	0	--	0	--
SVR and PQN ^2^	0.075	*p* < 2.2 × 10^−16^	0.100	*p* < 2.2 × 10^−16^
Creatinine and Specific Gravity ^2,4^	0.040	*p* < 2.2 × 10^−16^	0.002	0.21
Creatinine and PQN ^2^	0.116	*p* < 2.2 × 10^−16^	0.103	*p* < 2.2 × 10^−16^
Specific gravity and PQN ^2^	0.075	*p* < 2.2 × 10^−16^	0.101	*p* < 2.2 × 10^−16^

^1^ Abbreviations: RPLC, reversed-phase liquid chromatography; HILIC, hydrophilic interaction chromatography; RSD, relative standard deviation; SVR, support vector regression; PQN, probabilistic quotient normalization; SG_ref_, reference specific gravity; SG, specific gravity; QC, quality control sample. ^2^ Raw data were processed using XCMS. SVR normalization was then applied to all datasets. Creatinine, specific gravity, and PQN adjustments were carried out after SVR normalization. ^3^ RSD mean differences are computed in the order listed, for example, RSD_raw_–RSD_SVR_, and expressed as percentages. ^4^ Because of the specific gravity normalization method in which SG_ref_ = SG for QC samples (see Section 4.6, Equation (1)), QC peak areas are the same as those normalized by SVR alone, and therefore there is no difference in the RSD when comparing these methods. Similarly, because QC peak areas are the same for specific gravity and SVR, comparisons between creatinine and SVR and creatinine and specific gravity result in the same mean differences in terms of absolute value, and the same *p*-values.

**Table 3 metabolites-09-00198-t003:** Results of OPLS-DA analysis of variation associated with pre- and post-implantation status, and Wilcoxon paired signed-rank test of changes in peak area from pre- to post-implantation.^1^

Normalization Approach	RPLC Data					HILIC Data				
	Features, *n* ^3^	R^2^X	R^2^Y	Q^2^	VIP > 1 and *q* < 0.05, n ^4^	Features, *n* ^3^	R^2^X	R^2^Y	Q^2^	VIP > 1 and *q* < 0.05, n ^4^
Raw ^2^	9816	0.18	0.93	0.66	1303	8271	0.25	0.87	0.51	1385
SVR ^2^	12,023	0.13	0.95	0.62	1425	15,158	0.22	0.90	0.49	2161
Creatinine ^2^	11,744	0.36	0.82	0.58	1589	15,104	0.37	0.75	0.27	1491
Specific Gravity ^2^	12,023	0.37	0.86	0.62	1591	15,158	0.20	0.87	0.53	2358
PQN^2^	12,667	0.18	0.94	0.69	1551	18,106	0.25	0.91	0.52	2578

^1^ Abbreviations: OPLS-DA, orthogonal partial least-squares discriminant analysis; RPLC, reversed-phase liquid chromatography; HILIC, hydrophilic interaction chromatography; VIP, variable importance in projection; SVR, support vector regression; PQN, probabilistic quotient normalization. ^2^ Raw data were processed using XCMS. SVR normalization was then applied to all datasets. Creatinine, specific gravity, and PQN normalizations were then carried out after SVR normalization. ^3^ Only those features with RSD < 0.3 were included in this analysis. ^4^ Number of discriminant features with VIP > 1 based on OPLS-DA analysis, and *q* < 0.05 based on the Wilcoxon paired signed-rank test.

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
