# Peer review of "Normalizing Untargeted Periconceptional Urinary Metabolomics Data: A Comparison of Approaches"

_metabolites, 2019, doi:10.3390/metabo9100198_

Round 1
Reviewer 1 Report
Rosen and co-authors present here a concise a well written manuscript. The study aims to compare different normalisation approaches of urine metabolomics data around a specific time window: the periconceptional period. Using urine samples collected on 45 volunteers pre and post conception, they conclude that creatinine (the gold standard for urine metabolomics normalisation) may be less efficient and accurate than specific gravity measurements. The work presented here is of wide interest for the metabolomics community. A few issues and comments should be addressed before publication.
In the results section (table 1) the comparison of the methods should be validated by appropriate statistical tests and a P values should be provided to assess the significance of the findings Figure 1 and 2, the panel numbering should be clearer The conclusion finishes on the limitations of the study. Although all the points raised by the authors are valid and should therefore be mentioned, a final sentence/paragraph concluding on the significance of the findings and the main take home message would greatly improve the conclusion. In the material and method section, it would be interesting to include the rational of why the authors decided to pool samples from the chosen time points.
Reviewer 2 Report
The introduction of the paper is well written and it shows the necessity of the standardised normalization method for urine untargeted metabolomics. Comparing pre- and post-analytical normalization is a good idea and I agree with authors that when it comes to the large cohort studies (thousands of samples) the post-analytical is a good solution. Regarding the rest of the publication I have comments and suggestions:
SVR in this manuscript is used for signal-batch (S/B) correction, applied to all data sets before normalisation. This has to be rephrased in the paper and clarified. Perhaps justification, why SVR was chosen for S/B correction, is needed as well.
It can be argued if normalisation of raw data using creatinine, specific gravity and PQN has to be carried out before S/B correction and not after.
PCA (Fig 1 and 2) plots are not informative at all regarding effects of normalisation to the data structure. Some statistical testing, univariate or multivariate is needed to show the effect of normalisation; As authors claim in the manuscript SVR correction remove any technical noise from the data, so observing trends of QC samples are useless for normalisation method after SVR.
Against authors claims in results and discussion chapter, results from Table 1 indicate that SVR + PQN outperforms SVR + specific gravity method. Table 1 shows a different initial number of features despite the XCMS processing was done only once.
Even if the blank samples were acquired there is no mentioning in the publication how or if they were used during the sample processing.
I would suggest using a QC filter for features to get rid of the peaks which are a random end not sample related.
General comment: Results show in the current manuscript don't provide enough evidence that use of specific gravity measurement, which needs additional bioanalytical measurement, is outperforming PQN method.
Round 2
Reviewer 2 Report
Response to a point 3: Thank you for providing additional data for comparison of normalisation of the data. The data are more accurate now. Despite the citations, I still disagree that visual inspection of QC is alright. In my opinion, everything should be supported by numbers not just by a subjective observation.
Response to a point 5: I don’t see how PQN leads falsely to low RSD, it is just how the PQN normalisation works (out of all other normalisation methods CLR, ILR, total sum normalisation, works with least errors) and to eliminate the analytical variance from the instruments the peak areas of QC should be identical after correction. But this is for a longer discussion and not in the scope of this paper…
Response to a point 8: The QC filter is usually used to remove features which are present in at least 90% of QC samples to remove variable noise features. It is usually applied before or/and together with RSD filter. Obviously the threshold might change depending on the DOE.
Response to a point 9: In my opinion, the conclusions are more supported by the data now. I agree that specific gravity and PQN outperforms creatinine normalisation.